

# Deep-learned Top Tagging with a Lorentz Layer

Anja Butter[1], Gregor Kasieczka[2], Tilman Plehn[1⋆] and Michael Russell[1,3]

**1** Institut für Theoretische Physik, Universität Heidelberg, Germany
**2** Institute for Particle Physics, ETH Zürich, Switzerland
**3** School of Physics and Astronomy, University of Glasgow, Scotland

⋆ plehn@uni-heidelberg.de

## Abstract

We introduce a new and highly efficient tagger for hadronically decaying top quarks, based on a deep neural network working with Lorentz vectors and the Minkowski metric. With its novel machine learning setup and architecture it allows us to identify boosted top quarks not only from calorimeter towers, but also including tracking information. We show how the performance of our tagger compares with QCD-inspired and image-recognition approaches and find that it significantly increases the performance for strongly boosted top quarks.

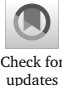
## Content

## 1 Introduction

The classification of hadronic objects has become the main driving force behind machine learning techniques in LHC physics. The task is to identify the partonic nature of large-area jets or

fat jets. Such jets occur for instance in boosted hadronic decays of Higgs bosons [1], weak gauge bosons [2–4], or top quarks [5–22].

A widely debated, central question is how we can analyze these jet substructure patterns using a range of machine learning techniques. An early example were wavelets, describing patterns of hadronic weak boson decays [23, 24]. The most frequently used approach is image recognition applied to calorimeter entries in the azimuthal angle vs rapidity plane, so-called jet images. They can be used to search for hadronic decays of weak bosons [25–29] or top quarks [30, 31], or to distinguish quark-like from gluon-like jets [32]. Another approach is inspired by natural language recognition, applied to decays of weak bosons [33].

Top taggers inspired by image recognition rely on convolutional networks (CNN) [31, 34], which work well for numbers of pixels small enough to be analyzed by the network. We have shown that they can outperform multi-variate QCD-based taggers, but also that the CNN learns all the appropriate sub-jet patterns [31]. A major problem arises when we include tracking information with its much better experimental resolution, leading to too many, too sparsely distributed pixels [32].

We propose a new approach to jet substructure using machine learning: rather than relying on analogies to image or natural language recognition we analyze the constituents of the fat jet directly, only using the Lorentz group and Minkowski space-time. For our DEEPTOPLOLA tagger we introduce a Combination layer (CoLa) together with a Lorentz layer (LoLa) and two fully connected layers forming a novel deep neural network (DNN) architecture. In the standard setup the input 4-momenta correspond to calorimeter towers [35]. However, unlike other approaches the DEEPTOPLOLA tagger can be extended to include tracking information and particle flow objects with their full experimental resolution in a technically trivial way.

This flexible setup allows us to study how much performance gain tracking information actually gives. Moreover, it means that DEEPTOPLOLA can be immediately included in state-of-the art ATLAS and CMS analyses and can be combined with $b$-tagging.

In this letter we first introduce our new machine learning setup. Using standard fat jets from hadronic top decays we compare its performance to multivariate QCD-inspired tagging and an image-based convolutional network [31]. We then extend the tagger to include particle flow information and estimate the performance gain compared to calorimeter information for mildly boosted and strongly boosted top quarks.

## 2 Tagger

The basic constituents entering any subjet analysis are a set of $N$ measured 4-vectors sorted by $p_T$, for example organized as the matrix

$$(k_{\mu,i}) = \begin{pmatrix} k_{0,1} & k_{0,2} & \cdots & k_{0,N} \\ k_{1,1} & k_{1,2} & \cdots & k_{1,N} \\ k_{2,1} & k_{2,2} & \cdots & k_{2,N} \\ k_{3,1} & k_{3,2} & \cdots & k_{3,N} \end{pmatrix} . \tag{1}$$

We show a typical jet image for a hadronic top decay in Fig. 1, indicating that the calorimeters entries of a typical top decay form a sparsely filled image.

### 2.1 Combination layer

Our tagger consists of two physics-inspired modules. As a first step, we multiply the 4-vectors from Eq.(1) with a matrix $C_{ij}$. Inspired by the treatment of jet clustering in the Qjets ap-

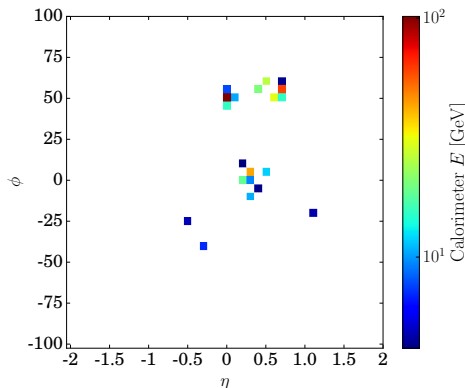

Figure 1: Jet image illustrating a signal event, showing 20 4-vectors $k_{\mu,i}$ with an energy threshold $k_0 > 1$ GeV on the calorimeter level.

proach [36] this defines our Combination layer

$$k_{\mu,i} \xrightarrow{\text{CoLa}} \tilde{k}_{\mu,j} = k_{\mu,i} \, C_{ij} \, . \tag{2}$$

It returns $M$ 4-vectors $\tilde{k}_j$, so $i = 1 \ldots N$ and $j = 1 \ldots M$. From many top tagging tests we known that an efficient tagger needs to find the mass drops associated with the top decay and the $W$ decay [13, 14, 31]. For illustration purposes, we look at the two corresponding on-shell conditions in our framework,

$$\begin{aligned}
\tilde{k}_{\mu,1}^2 &= (k_{\mu,1} + k_{\mu,2} + k_{\mu,3})^2 = m_t^2 \\
\tilde{k}_{\mu,2}^2 &= (k_{\mu,1} + k_{\mu,2})^2 = m_W^2 \, .
\end{aligned} \tag{3}$$

They correspond to non-zero entries

$$C_{11} = C_{21} = C_{31} \quad \text{and} \quad C_{12} = C_{22} \, . \tag{4}$$

In general, the CoLa matrix in our neural network has the trainable form

$$C = \begin{pmatrix}
1 & 0 & \cdots & 0 & C_{1,N+2} & \cdots & C_{1,M} \\
0 & 1 & & \vdots & C_{2,N+2} & \cdots & C_{2,M} \\
\vdots & \vdots & \ddots & 0 & \vdots & & \vdots \\
0 & 0 & \cdots & 1 & C_{N,N+2} & \cdots & C_{N,M}
\end{pmatrix} \, . \tag{5}$$

It guarantees that the set of $M$ 4-momenta $\tilde{k}_j$ includes

1. each original momentum $k_i$;
2. a trainable set of $M - N$ linear combinations.

These $\tilde{k}_j$ will be analyzed by a DNN. While one could use advanced pre-processing beyond some kind of ordering of the input 4-momenta, our earlier study [31] suggests that this is not necessary. For our numerical study we vary $N$ according to physics scenario. We use 15 trainable combinations, or $M = 15 + N$, where we have checked that changing $M$ has no effect.

## 2.2 Lorentz layer

From fundamental theory we know that the relevant distance measure between two substructure objects is the Minkowski metric. We use it to construct a weight function which makes it

easier for the DNN to learn the underlying features.[*] The Lorentz layer as the second part of the DNN first transforms the $M$ 4-vectors $\tilde{k}_j$ into the same number of measurement-motivated objects $\hat{k}_j$,

$$\tilde{k}_j \xrightarrow{\text{LoLa}} \hat{k}_j = \begin{pmatrix} m^2(\tilde{k}_j) \\ p_T(\tilde{k}_j) \\ w_{jm}^{(E)} E(\tilde{k}_m) \\ w_{jm}^{(d)} d_{jm}^2 \end{pmatrix} . \tag{6}$$

The four entries illustrate different structures we can include in this Lorentz layer. The first two $\hat{k}_j$ map individual $\tilde{k}_j$ onto their invariant mass and transverse momentum. The invariant mass entry corresponds to the illustration in Eq.(4). The third entry constructs a linear combination of all energies, with a trainable vector of weights $w_{jm}^{(E)}$ with $m = 1 \dots M$. Different values of $j$ give us a set of $M$ copies of this linear combination. The fourth entry combines all $\tilde{k}_m$ with a fixed $\tilde{k}_j$, including a trainable vector of weights $w_{jm}^{(d)}$. We can either sum over or minimize over the internal index $m$, always keeping the external index $j$ fixed. For the third entry with the trainable weights $w_{jm}^{(E)}$ we choose the sum over the internal index. For the last entry with the weights $w_{jm}^{(d)}$ we improve the performance of the network by including four copies with independently trainable weights. Two of these copies sum over the internal index and two of then minimize over it.

We have checked that neither the exact composition of the $\hat{k}_j$ nor the number of entries in Eq.(6) have an effect on the performance of our tagger. What is important is that we combine the invariant mass with an energy or transverse momentum and include the trainable weights. The first and last entries in Eq.(6) explicitly use the Minkowski distance,

$$d_{jm}^2 = (\tilde{k}_j - \tilde{k}_m)_\mu \, g^{\mu\nu} \, (\tilde{k}_j - \tilde{k}_m)_\nu . \tag{7}$$

The LoLa objects $\hat{k}_j$ are the input of the DNN. One can think of them as a rotation in the observable space, making the relevant information more accessible to the neural network, so the LoLa should be loss-less, provided the truncation in the number of input 4-vectors and the selection in Eq.(6) is carefully tested. Finally, the combined set of trainable weights in Eq.(5) and in Eq.(6) is large and can most likely be reduced for a given application. To maintain the general structure of our approach we decide to not apply this optimization.

## 3   Performance

For any proposed new analysis tool, a realistic and convincing comparison with the state-of-the-art tools is crucial. For our DEEPTOPLOLA tagger we compare its performance with a QCD-inspired top tagger and with an image-based top tagger, both working on calorimeter entries.

For our comparison we simulate a hadronic $t\bar{t}$ sample and a QCD di-jet sample with PYTHIA8.2.15 [37] for the 14 TeV LHC [38]. We ignore multi-parton interactions and in particular pile-up, which can eventually be removed [39–41]. Moreover, we assume that our top tagger can be trained on a pure sample of lepton-hadron top pair events with an identified leptonic top decay.

All events are passed through the fast detector simulation DELPHES3.3.2 [42], with calorimeter towers of size $\Delta\eta \times \Delta\phi = 0.1 \times 5°$ and an energy threshold of 1 GeV. We cluster these

---

[*]We are grateful to Johann Brehmer for pointing out that this approach limits us to fat jets far from black holes.

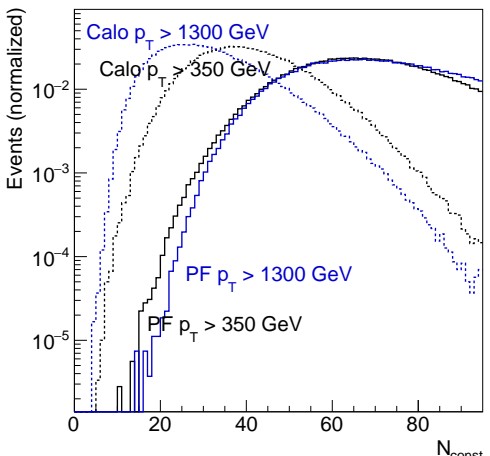
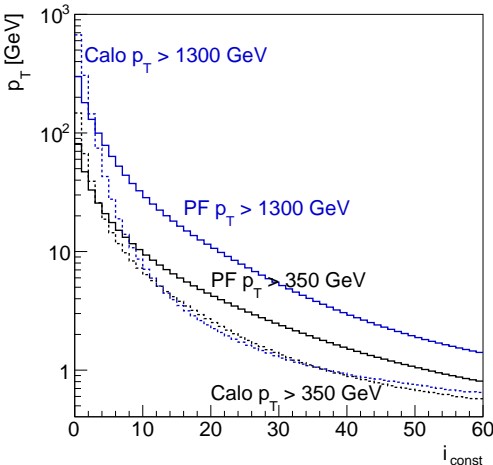

Figure 2: Number of constituents (left) and mean of the transverse momentum (right) of the ranked constituents available as 4-vectors in Eq.(1). We show 4-vectors for the top signal from calorimeter cells or jet images (dashed) and from calorimeter and tracker information combined through particle flow (solid).

towers with FASTJET3.1.3 [43, 44] to anti-$k_T$ [45] jets with $R = 1.5$. This defines a smooth outer shape and a jet area of the fat jet. The fat jets have to fulfill $|\eta_\text{fat}| < 1.0$, to guarantee that they are entirely in the central part of the detector and to justify our calorimeter tower size. For signal events, we require that the fat jet can be associated with a true top quark within $\Delta R < 1.2$. Unlike in our earlier study we do not re-cluster the anti-$k_T$ jet constituents, because we eventually include tracking information and do not focus on a comparison with QCD-inspired taggers [31].

## 3.1 Calorimeter

We consider the two standard ranges, moderately boosted tops available in Standard Model processes and highly boosted tops in resonance searches,

$$p_{T,\text{fat}} = 350 \dots 450 \text{ GeV}$$
$$p_{T,\text{fat}} = 1300 \dots 1400 \text{ GeV} . \tag{8}$$

In Fig. 2 we show the number of calorimeter-based 4-vectors $k_{\mu,i}$ as well as their ordered mean transverse momentum for the soft and hard fat jet selections of Eq.(8). For the soft and hard selections we have tested values $N = 10 \dots 60$ and find the using the leading $N = 40$ calorimeter constituents completely saturates the tagging performance. The remaining entries will typically be much softer than the top decay products and hence carry little signal or background information from the hard process.

For the softer fat jets we use 180,000 signal and 180,000 background events to train the network, 60,000 events each for tests during training, and 60,000 events each to estimate the performance. For technical reason the harder fat jets rely on a 10% smaller sample.

The network includes the CoLa, the LoLa, and two fully connected hidden layers, one with 100 and one with 50 nodes. It is trained using KERAS [46] with the THEANO [47] back-end, the ADAM optimizer, and a learning rate of 0.001. Training terminates either after 200 epochs or when the performance on the test sample does not improve for five epochs, typically after several tens of epochs. [†] We independently train five copies of the network, and compare their

---

[†]Using this setup, the training for the softer fat jets takes less than 15 minutes in total on a Tesla K80 using a p2.xlarge computing instance on Amazon Web Services.

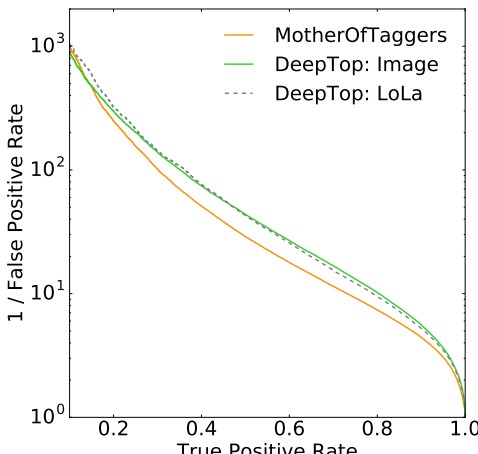

Figure 3: ROC curve for the new DEEPTOPLOLA tagger, compared to the QCD-inspired MOTHEROFTAGGERS and the image-based DEEPTOP tagger [31]. In all cases we only use calorimeter information for soft fat jets, $p_{T,\text{fat}} = 350 \ldots 450$ GeV.

performances on the independent validation sample.

Because of a long history of tests and applications on data, top taggers are especially useful to establish the performance of machine learning tools. In Fig. 3 we compare our DEEPTO­PLOLA tagger to earlier benchmarks for the softer of the two selections in Eq.(8): a BDT of a large number of QCD-inspired observables and the image-based DEEPTOP tagger [31]. The QCD-inspired MOTHEROFTAGGERS consists of a boosted decision tree which includes a large, relatively well-understood set of observables, which can be linked to a systematic approach to including sub-jet correlations [48]. It includes the HEPTOPTAGGER mass drop algorithm [14] with an optimal choice of jet size [17], different jet masses including SoftDrop [49], as well as N-subjettiness [50]. As long as we only include calorimeter information we cannot expect the new method to significantly improve over these two approaches. On the other hand, the num­ber of weights (inputs) of the LoLa-based DNN are lower by a factor of three to eight (ten to twenty) than what is used by the reference convolutional network. The proposed architecture is simpler, more flexible and physics-motivated but easily matches the convolutional network approach.

## 3.2 Learning the Minkowski metric

A technical challenge related to the Minkowski metric is that it combines two different features: two subjets are Minkowski-close if they are collinear or when one of them is soft ($k_{i,0} \to 0$). Because these two scenarios correspond to different, but possibly overlapping phase space regions, they are hard to learn for a DNN.

To see how our DEEPTOPLOLA tagger deals with this problem and to test what kind of structures drive the network output, we turn the problem around and ask the question if the Minkowski metric is really the feature distinguishing top decays and QCD jets. To this end, we define the invariant mass $m(\tilde{k}_j)$ and the distance $d_{jm}^2$ in Eq.(6) with a trainable diagonal metric. After applying a global normalization we find

$$g = \text{diag}(0.99 \pm 0.02 - 1.01 \pm 0.01, -1.01 \pm 0.02, -0.99 \pm 0.02), \qquad (9)$$

where the errors are given by five independently trained copies. It is crucial for our physics understanding [48] that the distinguishing power of the DEEPTOPLOLA tagger is indeed the

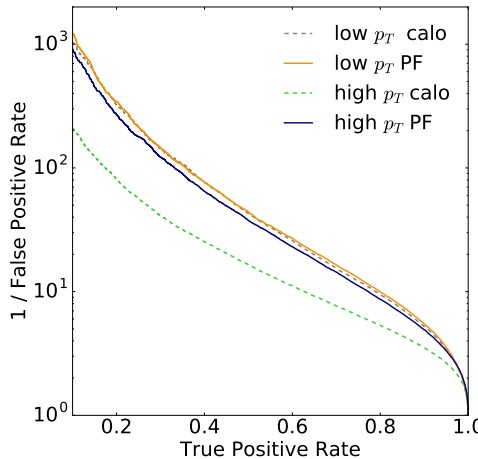

Figure 4: ROC curve for the new DEEPTOPLOLA tagger operating on particle flow objects, compared to the its performance operating on calorimeter objects.

same mass drop [1] that drives many QCD-based top taggers [13, 14] and the image-based top tagger, as shown in detail in Ref. [31].

## 3.3 Calorimeter and tracking

A standard criticism of the jet image approach is that the pixelled image removes information from the original jet. For the calorimeter information alone this is not the case, because the image pixels are given by the calorimeter resolution. However, this identification is not possible for tracking information, because the tracking resolution of ATLAS and CMS is much finer than a jet image can realistically resolve [32]. This makes it hard to in general extend jet images to particle flow objects and to reliably determine how much performance can be gained through tracking information.

In contrast, for our LoLa-based approach this extension to particle flow constituents is straightforward: instead of defining one constituent or 4-vector per calorimeter cell we use all objects defined by the DELPHES3 particle flow algorithm in the same $p_{T,\text{fat}}$ range as in Eq.(8). The fat jet constituents at the particle flow level are different from the calorimeter case, which implies that for the same $p_{T,\text{fat}}$ range the underlying top quarks are around 5% softer for fat jets based on particle flow objects. Nevertheless, defining the signal and background events using Eq.(8) still is the best choice.

In Fig. 2 we show the number of constituents for the calorimeter-level and the particle flow approaches. The latter offers not only considerably more objects, the corresponding 4-momenta are also measured more precisely. We also show the mean transverse momentum for each of these constituents, indicating that the larger number of particle flow objects at least in part arises from splitting harder calorimeter entries into several objects at higher resolution. For our DEEPTOPLOLA tagger Fig. 2 implies that we could include more particle flow objects than calorimeter objects in Eq.(1). Again, we use $N = 40$ and confirm that an increase to $N = 60$ has no measurable effect on the performance.

Searching for possible improvements to our tagger, we first check that indeed the top quark kinematics are more precisely measured by the particle flow objects. However, the observed 5% improvement, for example in the resolution of the top transverse momentum, is unlikely to significantly improve our analysis.

In Fig. 4 we confirm that using the same neural network for calorimeter and particle flow

objects gives hardly any improvement for moderately boosted tops with $p_{T,\text{fat}} = 350 \dots 450$ GeV. The situation changes when we train and test our tagger at larger transverse momenta, $p_{T,\text{fat}} = 1300 \dots 1400$ GeV. Here the calorimeter resolution is no longer sufficient to separate the substructures [51]. For a fixed signal efficiency the background rejection including particle flow increases by a factor of two to three.

## 4 Conclusions

Based on a deep neural network working on Lorentz vectors of jet constituents we have built the new, simple, and flexible DEEPTOPLOLA tagger. It includes a Combination layer mimicking QCD-inspired jet recombination, a Lorentz layer translating the 4-vectors into appropriate kinematic observables, and two fully connected layers. The 4-vector input is not limited to a single detector output but allows us to add more information about a subjet object in a straightforward manner.

We have compared the tagging performance to QCD-inspired taggers and to image-based convolutional network taggers using only calorimeter information for moderately boosted top quarks [31]. Figure 3 shows that the new tagger is competitive with either of these alternative approaches. Because we consider it crucial to control what machine learning methods actually exploit [48] we not only compared the DEEPTOPLOLA performance to an established QCD-inspired tagger [31], but also confirmed that the Minkowski metric related to a mass drop condition indeed drives the signal and background distinction.

Finally, we have used our tagger on particle flow objects, combining calorimeter and tracker information at their respective full experimental resolution. We have found that while for moderately boosted top quarks the performance gain from the tracker is negligible, it makes a big difference for strongly boosted top quarks.

The coverage of the full transverse momentum range and the possibility to include $b$-tagging through the tracking information should make the DEEPTOPLOLA tagger an excellent starting point to employ machine learning as the standard in ATLAS and CMS subjets analyses. It also opens a wide range of applications based on 4-vectors describing structures like for example matrix elements or phase space.

### Acknowledgements

First, we would like to thank Anke Biekötter for her help on tracking simulation. A.B acknowledges support form the *Heidelberg Graduate School for Fundamental Physics* and the DFG research training group *Particle Physics Beyond the Standard Model*. M.R. was supported by the European Union Marie Curie Research Training Network MCnetITN, under contract PITN-GA-2012-315877. Our work was supported by a grant from the Swiss National Supercomputing Centre (CSCS) under project D61.

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
