# Peer review of "Deep-learned Top Tagging with a Lorentz Layer"

_SciPost Physics, doi:SciPost Phys. 5, 028 (2018)_

## Round 2 · Referee Report · Anonymous (Referee 1) · 2018-2-19

Strengths

  • A new tagger for hadronically decaying top quarks is proposed
  • The technique makes used of deep learning techniques
  • It extends earlier work to include tracking on top of calorimeter towers
  • The technique can be deployed directly by the experiments now.

Weaknesses

  • The authors take a simplified situation with eg no pile-up added to the events. While the authors claim this can be removed, this is only possible to a certain level in the experiments, and it would have surely of interest to see if this tagger remains performant in the presence of pile-up, in particular since the pile-up a the LHC is expected to increase significantly in the next years.

Report

Deep learning techniques are a very promising field to assist analysers in recognising patterns in for example collision data in particle physics.
This paper discusses a novel tagger that has the potential to have a higher performance in particular for configurations where the decaying top quarks are boosted in the laboratory/detector system.

Requested changes

No big points but a cosmetics or additions

Figure 2 takes some time to understand. Perhaps the vertical access on the right figure should be labelled <pT> . Also N_const and I_const are not defined in the text I believe (or caption) although one can easily derive what they mean.

Can the authors comment on how the performance of this tagger could be affected by pile-up or at least point out the problems that can occur. There is
no request/need to actually perform a study with pile-up for this paper (can be in a future study)

For non experts: what is a learning rate of 0.001 correspond to? (page 6)

page 6: "the using the leading" something wrong

  • validity: high
  • significance: good
  • originality: high
  • clarity: good
  • formatting: excellent
  • grammar: excellent

Author:  Tilman Plehn  on 2018-04-23  [id 247]

(in reply to Report 1 on 2018-02-19)
Category:
answer to question

First of all, we would like to thank the referee for his/her helpful comments.

Figure 2 takes some time to understand. Perhaps the vertical access on the right
figure should be labelled <pT> . Also N_const and I_const are not defined in the
text I believe (or caption) although one can easily derive what they mean.

-> we added the definition to the text and slightly expanded the description.

Can the authors comment on how the performance of this tagger could be affected by
pile-up or at least point out the problems that can occur. There is
no request/need to actually perform a study with pile-up for this paper (can be in a
future study)

-> we would indeed prefer to postpone this question to a dedicated study, but we
added some ideas how pile-up removal could be combined with our tagger.

For non experts: what is a learning rate of 0.001 correspond to? (page 6)

-> we added a brief explanation.

page 6: "the using the leading" something wrong

-> changed.

---

## Round 2 · Referee Report · Anonymous (Referee 2) · 2018-3-15

Strengths

I could not identify strenghts, because the paper is not written very clearly. It uses a lot of jargon and is hard to follow.
In Figure 3, which shows the performance of the new tagger in comparison to a different already existing tagger, the improvements are not very impressive.

Weaknesses

1 - it has too much jargon 2 - I am not impressed by the perfomance improvements I see in figure 3. 3 - If I actually would wanted to try this new tagger out I would not be able to set it up, because the description of it is confusing. I have not understood it. I read the paper a few times now and still do not understand in detail how to set it up. 4- The novelty of the approach is not clear and what the advantages are.

Report

I am not sure which the audience this paper is aiming for. If it is meant for a general particle physicists it is not well written or understandable. If the paper is aiming for experts in machine learning techniques, it may be alright but I can not judge this because my expertise lies in jet substructure and standard methods to tag heavy objects.

Requested changes

Here are my detailed comment. I hope they will be helpful to address the issues I have raised above.

1) page 2, 3rd paragraph, last line: "....sparsely distributed pixels [21]." This is too vague. Can you please quantify this? And usually the pixel detectors have a lot of pixels which are very densely packed...compared to calorimeter cells they are not sparsely distributed. Hence, I do not really understand what you are meaning with this expression at the end of the sentences.

2) page 2, 4th paragraph: This is the paragraph where you are introducing your novel idea, hence a very important section in your paper. And it is, packed with jargon and does not convey the key message. What is novel about your tagger? Can you not write it without using expressions which the reader will not be able to understand at this stage of the paper. When you write you are analyzing the constituents only using the Loretnz group and Minkowski space-time, the English here is not really good. You mean you are using relationships or mathematical relationships based on the Lorentz grop and minkowski space-time formalism? I also do not understand why you are high-lighting that "...unlike other approaches the DeepTopLoLa tagger can be extended to include tracking information and particle flow objects with their full experimental resolution in a technically trivial way." Why? I do not see conceptually any problems to include tracking or PF information into any other NN algorithm.

3) page 2, 4th paragraph: please give a reference for particle flow objects.

4) Figure 1: Is this figure based on Monte Carlo simulation? Please state that in the caption. What level of simulation is this? Generator level? or reconstruction level? You also need to say that these 4-vectors in the phi-eta plane and also need to tell the reader in the caption that the z-scale is showing the energy of the jets.

5) Figure 1: is each of these squares in this figure a jet?

6) page 3, first sentence under Figure 1: "We show a typical....", Please check the English of this sentence. It does not read well after the "," something is wrong with it.

7) Section 2.1 is very confusing and I really do not understand your algorithm or what you have implemented.

8) Section 2.1, first paragraph, line 2 and 3: "...in the Qjets approach [25] ...." i think you should briefly describe what the Qjet approach is and how this matrix Cij is related to the Qjet approach. It is not fair to let the reader go back to reference [25] to find this out themselves.

9) Equation 2 tells me how I can compute the "higher level four vectors of the top and the W candidate based on the constitutent 4-vectors. But you do not give the reader any indication how you constrain the determination of the weights of the Cij matrix. What is this matrix trained to optimize?

10) Text below equation 2: I do not understand the meaning of the variable "M". If you have a di-top event you would have 2 top quark candidates and 2 W candidates. hence this means M should be at most = 4. But on page 4, first paragraph you see that M can be 15 + N. ??? Why? Why 15 + N, hence what is N the number of? Could you please define what M is standing or counting and what N is counting??

11) Text below equation 2: You write "For our numerical study we vary N according to physics scenario." How many scenarios did you study? I thought just the semi-leptonically decaying ttbar scenario, no? And how does N depend on these different scenarios?

12) Equation 6: You are using a variable d_jm which you only introduce in equation 7. Can you re-order this and make sure that you introduce all variables before using them?

13) I do not understand equation 6. I am so confused, that I am not even able to make a suggestion how to improve it. What is this equation doing? How would I implent it? Is this a new vector??

14) Figure 2: What is "i_const"? Could you please also use a legend instead of putting the labels next to the graphs?

15) What are epochs? Please define.

16) Page 6, second to last paragraph: You write "We independently train five copies of the network, and compare..." What is the difference between these copies? Do you use different training samples? or different seeds for the weights? this is not clear to me.

17) Section 3.2: I do not understand why what you are describing and your results in equation 9 show that your tagger is distinguishing top decays and QCD jets.

18) page 7, last line: Are you sure that you are refencing the right reference?

19) page 8, second sentence: "The latter offers not only ....the corresponding 4-vectors are also measured more precisely." Where do I see this? i mean that PF jets have a better resolution.

  • validity: low
  • significance: low
  • originality: low
  • clarity: poor
  • formatting: below threshold
  • grammar: reasonable

Author:  Tilman Plehn  on 2018-04-23  [id 248]

(in reply to Report 2 on 2018-03-15)
Category:
answer to question

1) page 2, 3rd paragraph, last line: "....sparsely distributed pixels [21]." This is too vague. Can you please quantify this? And usually the pixel detectors have a lot of pixels which are very densely packed...compared to calorimeter cells they are not sparsely distributed. Hence, I do not really understand what you are meaning with this expression at the end of the sentences.

-> we added 'active' pixels, to indicate that this statement refers to the pixes used by the tagger.

2) page 2, 4th paragraph: This is the paragraph where you are introducing your novel idea, hence a very important section in your paper. And it is, packed with jargon and does not convey the key message. What is novel about your tagger? Can you not write it without using expressions which the reader will not be able to understand at this stage of the paper. When you write you are analyzing the constituents only using the Loretnz group and Minkowski space-time, the English here is not really good. You mean you are using relationships or mathematical relationships based on the Lorentz grop and minkowski space-time formalism? I also do not understand why you are high-lighting that "...unlike other approaches the DeepTopLoLa tagger can be extended to include tracking information and particle flow objects with their full experimental resolution in a technically trivial way." Why? I do not see conceptually any problems to include tracking or PF information into any other NN algorithm.

-> we changed the sentence about the Minkowski metric and added a sentence explaining the problems in combining calorimeter and tracking images with very different resolution.

3) page 2, 4th paragraph: please give a reference for particle flow objects.

-> included.

4) Figure 1: Is this figure based on Monte Carlo simulation? Please state that in the caption. What level of simulation is this? Generator level? or reconstruction level? You also need to say that these 4-vectors in the phi-eta plane and also need to tell the reader in the caption that the z-scale is showing the energy of the jets.

-> our entire analysis is based on Monte Carlo simuation, but we have specified the details in the caption. Why we have to repeat the information from the axis labels is not immediately clear to us.

5) Figure 1: is each of these squares in this figure a jet?

-> obviously, those are not jets, but the jet constituents. We nevertheless added this to the caption.

6) page 3, first sentence under Figure 1: "We show a typical....", Please check the English of this sentence. It does not read well after the "," something is wrong with it.

-> we are not sure which part of that sentence 'does not read well'.

7) Section 2.1 is very confusing and I really do not understand your algorithm or what you have implemented.

-> admittedly, we are trying to keep this section very brief. We changed the end of Sec.2.1 to maybe make it more clear, but it is not clear which part of Eqs.(2-5) the referee is referring to.

8) Section 2.1, first paragraph, line 2 and 3: "...in the Qjets approach [25] ...." i think you should briefly describe what the Qjet approach is and how this matrix Cij is related to the Qjet approach. It is not fair to let the reader go back to reference [25] to find this out themselves.

-> Qjets is an established concept in subjet physics and not easily explained in one sentence. However, we added 'non-deterministic' to remind the reader why we mention it as an analogy.

9) Equation 2 tells me how I can compute the "higher level four vectors of the top and the W candidate based on the constitutent 4-vectors. But you do not give the reader any indication how you constrain the determination of the weights of the Cij matrix. What is this matrix trained to optimize?

-> we are not sure we understand this comment. The weights of the Cij matrix are part of our neural network, trained to tell top jets from QCD jets?

10) Text below equation 2: I do not understand the meaning of the variable "M". If you have a di-top event you would have 2 top quark candidates and 2 W candidates. hence this means M should be at most = 4. But on page 4, first paragraph you see that M can be 15 + N. ??? Why? Why 15 + N, hence what is N the number of? Could you please define what M is standing or counting and what N is counting??

-> N is the number of input constituent 4-vector defined next to Eq.(1) and M is defined with Eq.(2). We now repeat the definition of N when we first introduce M.

11) Text below equation 2: You write "For our numerical study we vary N according to physics scenario." How many scenarios did you study? I thought just the semi-leptonically decaying ttbar scenario, no? And how does N depend on these different scenarios?

-> we now specify in the text that we look at calorimeter vs particle flow and at moderate vs strong boost.

12) Equation 6: You are using a variable d_jm which you only introduce in equation 7. Can you re-order this and make sure that you introduce all variables before using them?

-> the definition is now where the referee suggested to put it.

13) I do not understand equation 6. I am so confused, that I am not even able to make a suggestion how to improve it. What is this equation doing? How would I implent it? Is this a new vector??

-> now we are confused. This vector khat are the variables which the network actually uses to separate top jets from QCD jets. This is stated in the very beginning of Sec.2.2.

14) Figure 2: What is "i_const"? Could you please also use a legend instead of putting the labels next to the graphs?

-> we now explain the labels in the text. Concerning the leyout, we believe that the plots are easily understandable in their current layout.

15) What are epochs? Please define.

-> while 'epoch' should be a well-known term in machine learning, we did add a brief reminder to the text, at the risk that expert readers will doubt our expertize.

16) Page 6, second to last paragraph: You write "We independently train five copies of the network, and compare..." What is the difference between these copies? Do you use different training samples? or different seeds for the weights? this is not clear to me.

-> Exactly, we use different seeds for initialising the weights

17) Section 3.2: I do not understand why what you are describing and your results in equation 9 show that your tagger is distinguishing top decays and QCD jets.

-> the main result of Sec.3.1 is that our tagger distinguishes top quarks from QCD jets. So this Sec.3.2 is really about understanding what drives the performance we have seen before.

18) page 7, last line: Are you sure that you are refencing the right reference?

-> we are not sure what the referee is referring to, every reference seems to look fine.

19) page 8, second sentence: "The latter offers not only ....the corresponding 4-vectors are also measured more precisely." Where do I see this? i mean that PF jets have a better resolution.

-> this is just referring to the experimental fact that PF objects have a better resolution. We do not show this explicitly in our analysis, only mention it in the introduction.

---

## Editorial Decision

published